# The Use of Post-Natal Skeleton Development as Sensitive Preclinical Model to Test the Quality of Alternative Protein Sources in the Diet

**DOI:** 10.3390/nu14183769

**Published:** 2022-09-13

**Authors:** Astar Shitrit-Tovli, Roni Sides, Rotem Kalev-Altman, Dana Meilich, Gal Becker, Svetlana Penn, Ron Shahar, Efrat Monsonego Ornan

**Affiliations:** 1Institute of Biochemistry and Nutrition, The Robert H. Smith Faculty of Agriculture, Food and Environment, The Hebrew University of Jerusalem, Rehovot 7610001, Israel; 2Koret School of Veterinary, The Robert H. Smith Faculty of Agriculture, Food and Environment, The Hebrew University of Jerusalem, Rehovot 7610001, Israel

**Keywords:** protein quality, bone development, soy protein, spirulina algae, chickpea protein, chickpea flour, insects’ protein

## Abstract

Dietary protein is necessary throughout all life stages. Adequate intake of protein during juvenile years is essential to enable appropriate synthesis of bone matrix and achieve the full peak bone mass (PBM). Due to socio-demographic changes, accompanied by environmental damage and ethical problems, a transition to the consumption of different and alternative protein sources in the human diet must occur. This transition requires the precise evaluation of protein quality. Here, we utilize a preclinical model of young rats during their post-natal developmental period to define the nutritive quality of a number of alternative protein sources (soy, spirulina, chickpea, and fly larvae) by their health impact on growth performance and skeletal development. We indicate that when restricted (10% of calories) not one of the tested alternative protein sources have succeeded in causing optimal growth, as compared to the referenced source, casein; yet fly larvae protein followed by chickpea flour were found to be superior to the rest. Growth-plate histology and µ-CT analyses demonstrated a number of changes in growth patterns and bone morphometric parameters. Bone mechanical testing, by three-point bending analyses, was sensitive in demonstrating the effect of the reduction in the amount of the dietary protein. Moreover, the rats’ weight and length, as well as their eating patterns, were found to reflect the proteins’ quality better than their amino acid composition. Hence, our study emphasizes the importance of evaluating protein as a whole food source, and suggests a new approach for this purpose.

## 1. Introduction

Protein is an essential element of a healthy diet, allowing the synthesis of the 25,000 proteins encoded within the human genome, as well as other nitrogenous compounds. Both form the body’s dynamic system of structural and functional elements that exchange nitrogen with the environment. Therefore, adequate intake of dietary protein is necessary throughout all stages of life, and especially during growth, pregnancy, and lactation [1].

The musculoskeletal system is one of the many organs that require a sufficient dietary protein intake [2]. Total bone protein is about 50% of bone volume and approximately one-third of its mass [3]. Consumption of exogenous amino acids in young adulthood is necessary to enable the appropriate synthesis of protein–bone matrix and achieve the full genetic potential for skeletal mass, the peak bone mass (PBM), when sexual maturation occurs [4].

The continuing consumption of the current conventional protein sources, such as meat, dairy, fish, and eggs, raises concern about sustainability and food security. Animal-based foods have higher greenhouse gas (GHG) emissions than plant-based foods [5]. Moreover, the increased demand for animal-based protein enhances the need to produce more animal feed, and this leads to the conversion of forests, wetlands, and natural grasslands into agricultural lands [6]. Hence, it is indispensable to find alternative, sustainable, and healthy protein sources.

Nowadays, the most prominent options to be provided as a substitute for conventional dietary proteins are legumes, algae, yeast, and insects [7]. Legumes contain about 21–25% crude protein (in dry basis) [8] and have a significant role in overcoming challenges associated with protein–energy malnutrition in developing countries [9]. Among legumes, soybean is prominent due to its higher ratios of essential amino acids. Protein digestibility-corrected amino acid scores (PDCAAS) for modern soy products such as soy isolate/concentrate are similar to the animal proteins’ scores [10]. Furthermore, dietary soy protein intake is associated with reduced adiposity, blood glucose, and insulin; and with improvements in lipid profile and insulin sensitivity [11,12]. Some of these effects are associated with the phytochemicals that soy contains, such as isoflavones. 

Another legume that contains isoflavones and is characterized by a good balance of amino acids, is the chickpea [13,14]. Due to its protein content and beneficial effects on human health, the chickpea is considered a potential alternative and sustainable plant-based protein and so its isolates have been gaining popularity. Several studies that examined treatments on ovariectomized (OVX) rats with soy and chickpea extracts revealed that the estrogen-like effects of isoflavones can prevent bone loss induced by ovarian hormone deficiency [15,16,17,18]. Furthermore, soy intake was reported to increase the secretion and the bioavailability of insulin-like growth factor I (IGF-I) [19], while chickpea extracts diminish RANKL levels, and increase its soluble decoy receptor, osteoprotegerin (OPG) [18,20].

Another potential protein source is algae [21]. One of the most popular microalgae is spirulina, blue-green cyanobacteria. Currently, spirulina is listed by the Food and Drug Administration (FDA) under the category Generally Recognized as Safe (GRAS) [22], and it is sold mainly as a dietary supplement [23]. However, it has also been promoted as “the food of the future” due to its protein content (60–70% of dry weight) [24] and essential amino acid composition. Spirulina contains phycocyanin, an antioxidative pigment with anti-inflammatory, neuroprotective, and hepatoprotective effects [25]. Additionally, it is an important source of essential fatty acids, including the rare γ-linolenic acid (18:3ω6, GLA) [26]. Regarding the skeletal system, a previous study revealed that adding 5 g/kg spirulina to a low-protein-low-micronutrients diet for growing rats improved their growth performances and bone quality [27].

Insects are part of the traditional diets of at least 2 billion people worldwide, particularly in parts of Asia, Africa, and South America [7]. There are more than 1900 edible insect species [28]. They have been identified as an alternative protein source for the Western world while the Food and Agriculture Organization (FAO) promotes their consumption as a healthy and environmentally friendly food source [28]. One of the potential insect species in the alternative proteins market is the Mediterranean fruit fly (*Ceratitis capitata*) larvae which is a good source of protein (about 67%), monounsaturated fats (4.1%), and minerals such as iron, phosphorus, calcium, and magnesium. So far, no studies have been performed regarding protein quality and the health effects of this protein source. 

In this present study, we aimed to determine which of the innovative protein sources—soy, chickpea, spirulina, and fly larvae—is good enough to be an adequate substitute for the conventional dietary proteins, in terms of their impact on growth performance. To do so, we established a sensitive preclinical model of young rats during their post-natal developmental period. Through our investigative approach, our study can help us to understand the effect of varied dietary proteins on skeletal system development, and, simultaneously, enable us to test the quality of different protein sources such as whole foods.

## 2. Materials and Methods

### 2.1. Animals

Three-week-old 56 Female Sprague Dawley (SD) rats after weaning were purchased from Harlan Laboratories (Rehovot, Israel) and housed in environmentally controlled conditions. All procedures were approved by the Hebrew University Animal Care Committee (permit number AG-20-16457-3). After 4 days of adaptation to a control chow diet, we randomly divided the rats into 7 groups, with 8 rats in each group. Two of the groups consumed casein as a standard protein source; while one (1) consumed the standard control diet (21% protein, Ctrl group), and the other (2) consumed the deficient control diet (10% protein, PD-Ctrl group). The five remaining experimental groups consumed protein deficient diets (10% protein), whereas the alternative protein source was either from (3) soy isolate (PD-Soy); (4) spirulina powder (PD-Spl); (5) chickpea isolate (PD-CP/I); (6) chickpea flour (PD-CP/F); or (7) fly larvae protein powder (PD-Fly). All rats had an *ad libitum* access to food and liquids.

At the end of the experiment, after 6 weeks, the rats were anesthetized with isoflurane and blood samples were collected and stored at −80 °C. Next, their femur, tibia, and spine were harvested and stored at −20 °C. The femurs and lumbar vertebras were manually cleaned up of soft tissue, wrapped in saline-soaked gauze, and stored until analysis (mechanical/micro-CT testing). Finally, tibias were fixed for histological analysis.

### 2.2. Diet Preparation and Composition

During the post-weaning period, rats from the Ctrl group were fed a diet based on the American Institute of Nutrition (AIN-93) recommendation that was formulated for the growth phase of rodents (Table 1): 18% fat, 61% carbohydrate, and 21% protein from a source of casein [29]. In addition, the rest of the groups were fed protein-deficient (PD) diets that were high in fat and low in protein (Table 1): 28% fat, 62% carbohydrate, and 10% protein. Each PD group was fed a different protein source, so the amount of the ingredients in each diet was adjusted and balanced according to the nutritional content of the protein source (Table 2). The entire diet was homogenized, shaped as dumplings, and frozen at −20 °C.

### 2.3. Histological Staining of Growth-Plate (GP) Sections

Safranin-O staining was used to examine the tibial growth plates (GPs). Tibia samples were fixed overnight in 4% paraformaldehyde (PFA, Sigma, St. Louis, MO, USA) at 4 °C followed by 3 weeks of decalcification in 0.5 M EDTA pH 7.4. Afterward, the samples were dehydrated by a 1-h wash with increased concentrations of ethanol (70%, 85%, 95%, and 2 × 100%), transferred into Histo-Clear (Bar-Naor, Israel) twice for 2 h each, and embedded in paraffin blocks. By using a Leica (Agentec, Yakum, Israel) microtome, we prepared 5 µm transverse tissue sections for histological staining [30,31].

The sections were deparaffinized by heating the slides, followed by two washes with xylene, rehydration by washes with decreasing ethanol concentrations (2 × 100%, 95%, 85%, and 70%), and a final wash with distilled water. Weigert’s iron hematoxylin solution, a fast green solution, and acetic acid were used for Safranin-O staining. The sections were dried and mounted with DPX mounting for histology. 

### 2.4. Imaging and Measurement of GPs

Stained transverse sections of tibiae were viewed by an eclipse E400 Nikon light microscopy with ×4, ×10, ×20, or ×40 objectives, using light filters. Images were taken by a high-resolution camera (Olympus DP 71) controlled by Cell A software (Olympus, Japan). The thickness of the entire GPs, the proliferative zone (PZ), and the hypertrophic zone (HZ) as well as the number of cells were measured using the Cell A software with a measuring tool feature. Measurements were performed on these sections from 4 different animals from each group. In each slide, 10 random locations throughout the GPs were selected and measured [32,33].

### 2.5. Bone Microarchitecture

The right femora were scanned using a Skyscan 1174 (Skyscan, Bruker, Belgium) X-ray computed micro-tomography device. Images were obtained at 50 kV X-ray tube voltage and 800 µA current, using a 0.25 mm aluminum filter, 4000 ms exposure time, and 15 µm optical resolution. For each specimen, a series of 900 projection images were obtained (a rotation step of 0.4°, averaging 2 frames, for a total 360° rotation). A stack of 2-D X-ray shadow projections was reconstructed to obtain images using NRecon software (Skyscan). Next, images were subjected to morphometric analysis using CTAn software (CT Analyser 1.13.5.1, Skyscan). Morphometric parameters were calculated as suggested by recent guidelines for bone microstructure assessment. To analyze the diaphyseal cortical region, 200 slices, centered at the mid diaphysis, equivalent to 2.764 mm, were chosen. Global grayscale threshold levels for the cortical region were between 72–255. For the trabecular region, a total of 100 slices, equivalent to 1.382 mm of the bone, were selected, and adaptive grayscale threshold levels between 72 and 255 were used. Two phantoms with known density (0.25 and 0.75 g/cm^3^) were scanned under the same conditions as the femora samples allowing us to measure the cortical diaphysis BMD (bone mineral density); quantifications were carried out using CTAn software [34,35]

The 3rd–5th lumbar vertebrae were scanned and analyzed as well. The filter was changed to 0.5 mm, spatial resolution was 21.6 µm, and the total rotation was 180°, except that all other parameters were identical to the femoral scans. The region chosen for the analysis was manually selected and consisted of 100 slices of the 5th vertebra starting from the proximal end-plate. Adaptive grayscale threshold levels between 62 and 255 were selected for the analysis of the trabecular vertebral region. The length of the femora and the 3rd–5th segments of the lumbar vertebrae were measured using the Micro-CT device prior to the scans. 

### 2.6. Three-Point Bending for Bone Mechanical Analysis

The left femora were tested using an Instron mechanical tester (Model 3345). Each bone was placed within a custom-built saline-containing testing chamber and on two supports having rounded profiles (2 mm in diameter), so that the supports were in touch with the posterior aspect of the diaphysis. The distance between the stationary supports was set to 10 mm to ensure that the relatively tubular portion of the mid-diaphysis rested on these supports. A pronged loading device was applied to the anterior surface of the bones, precisely in the middle between the two supports. First, an initial preload of 0.1 N was applied to hold the bone in place; following that, the prong was advanced at a constant rate of 600 µm/min, loaded up to the fracture point, identified by a sudden >20% decrease in load [34]. Force-displacement data were collected by the Instron software BlueHill (version 2.0, Instron Corporation, Norwood, MA, USA) at 10 Hz. The resulting force–displacement curves were used to calculate bone stiffness, bone yield point, load of fracture, maximal load, and the area under the curve was measured to calculate the total energy to fracture (E to F) [36]. Moreover, a stress–strain curve was calculated in order to determine the stiffness of the material excluding bone geometry, the Young’s modulus [37].

### 2.7. Serum Biochemistry, Diets’ Macronutrients, and Amino Acids Analyses

Blood samples were collected and centrifuged at the time of the sacrifice. A complete biochemical analysis was performed by the Veterinary Teaching Hospital (Bet Dagan, Israel).

Macronutrient analysis of the protein sources was performed by AMINOLAB analytical laboratory (Ness Ziona, Israel). Protein content was determined by the Kjeldahl method. Diets were hydrolyzed with concentrated sulfuric acid (H_2_SO_4_) at 420 °C. After cooling, H_2_O was added to the hydrolysates before neutralization and titration. The amount of total nitrogen was multiplied with the traditional conversion factor of 6.25. In addition, amino acid analysis was performed by MILOUDA & MIGAL analytical laboratories (Oshrat, Israel). Protein sources were hydrolyzed and the separated amino acids were quantitatively determined by using ion-exchange chromatography with an automatic analyzer.

### 2.8. Heat Mapping

All relevant data of each map (growth and eating patterns, bone analyses and serum biochemistry, and the protein’s amino acid content) was collected in one table. Values underwent normalization by scaling between 0 to 1 by using Excel software. Finally, heat maps were created by using Graphpad prism 9 software (GraphPad Software, San Diego, CA, USA).

### 2.9. Statistical Analysis

All data are expressed as mean ± SD. The normality of data and the significance of differences between groups were determined using JMP 14.0.0 Statistical Discovery Software (SAS Institute 2000) by Bartlett’s test and one-way analysis of variance. Differences between groups were further evaluated by a Tukey–Kramer HSD test and considered significant at *p* < 0.05.

## 3. Results

In order to evaluate the influence of different sources of protein on post-natal skeletal development, a 6-week-long experiment was conducted on SD rats after weaning. The rats were sacrificed at 9 weeks old. This time frame was selected with the intention to mimic the human growth period up to sexual maturity [38]. During this accelerated growth period, the rats were fed either a normal diet (20% protein) or a protein deficient (PD) diet (10% protein), while using different protein sources. The limited amount of protein ensures that the majority of the dietary amino acids have been utilized for body anabolism instead of being oxidized for metabolic energy. 

### 3.1. The Effect of the Alternative Protein Sources on Growth Patterns

The effect of the experimental diets on growth was first determined in terms of body weight (g), representing body growth rate including energy balance, and length from tip to nose (cm), as an indicator for longitudinal bone growth.

With the aim of evaluating the effect of PD diet on growth performances, a comparison of Ctrl groups (diets based on casein) has been made. Interestingly, both groups had shown comparable values of absolute weight gain (Figure 1A), along with a similar increase in body length throughout the whole period (Figure 1B). Correspondingly, average lengths of the right femora and the 3rd–5th lumbar vertebrae, which were scanned and measured by micro-CT, showed close values as well (Figure 1C,D). 

Next, we made a comparison between the groups that were fed the PD diet (10% protein), while using different protein sources. Although the rats were fed the same nutrient ratios, they demonstrated diverse growth patterns. Compared with a total of 150.6 g weight gain in the PD-Ctrl group, PD-Fly and PD-CP/F added only 102.3 g and 82.8 g, respectively; whereas PD-Spl, PD-CP/I, and PD-Soy gained about one-third of PD-Ctrl’s total weight gain (Figure 1E; Appendix A for statistical differences). This groups’ hierarchy had also shown in body length, and bone longitudinal growth (Figure 1F–H; Appendix A).

Since longitudinal bone growth originates from the growth plate (GP) [39], and bone elongation was diverse for most of the groups, we performed Safranin-O staining on the tibiae GP. This staining dyes the cartilage in a noticeable red-orange (Figure 2A). In general, all groups showed normal GP morphology. However, measurements of GP width and chondrocyte numbers revealed differences between the groups that were fed spirulina, chickpea isolate, and soy isolate. Compared with the PD-Ctrl group, PD-Spl demonstrated a wider proliferative zone (PZ) and a larger number of cells in this area. On the other hand, the Spirulina group’s hypertrophic zone (HZ) was comparable in its width to that of the PD-Ctrl, whereas the soy and the chickpea isolate’s diets led to smaller HZ (Figure 2B,C).

### 3.2. The Effect of the Alternative Protein Sources on Food Intake and Its Utilization for Growth

Throughout the experiment, food intake was measured for calculating energy and protein consumption, in order to evaluate the caloric input and the protein contribution to growth.

During the whole experiment, all groups exhibited energy intake that was compatible with their body weight: Ctrl groups had the highest energy intake, then PD-Fly and PD-CP/F groups, afterward PD-CP/I and PD-Spl, and then PD-Soy (Figure 3A). However, the evaluation of the first week of the experiment, when the rats had the same body weight, showed a different picture. PD-Ctrl consumed about 16% more calories than the Ctrl group but gained only about two-thirds of the Ctrl’s weight gain. PD-Spl consumed approximately 91% of the calories that PD-Ctrl consumed but gained only 37% of the PD-Ctrl’s weight gain. Furthermore, the comparison between PD-CP/I and PD-CP/F showed that both groups had equal energy intake, yet the rats that were fed chickpea flour gained significantly more weight (Figure 3B,C).

In an attempt to determine dissimilarities in protein quality independently from food intake, we calculated the protein efficiency ratio (PER)—the body weight gain per gram of consumed protein. PER value for casein was found to be the highest. Among the alternative protein sources, fly larva protein was the most qualitative, followed by chickpea flour, whereas spirulina powder, soy isolate, and chickpea isolate presented the poorest scores (Figure 3D).

### 3.3. The Effect of the Alternative Protein Sources on Serum Biochemistry Parameters

Next, we evaluated the serum biochemical profile and functional capacity of several critical organs and systems. Liver function was tested by liver enzymes and bilirubin levels and was found to be normal in all groups. Renal functions, tested by urea and creatinine blood levels, were all in the normal ranges. Nutritional status and protein utilization were tested by albumin and total protein (TP) blood levels. All alternative protein groups demonstrated low values of blood TP but normal values of albumin. No differences were found in creatinine, a biomarker for muscle catabolism, and urea serum concentrations, an indicator for protein utilization. Additionally, glucose values did not differ and were within the reference range or above it, since it is a non-fasting assessment, and lipid profile, cholesterol, and triglycerides (TG) blood concentrations, had standard values.

Bone health was tested by calcium and phosphorous levels, the major mineral constituents of bone. Phosphorous concentrations of PD-Ctrl, PD-Soy, and PD-Spl groups were below the normal range, consequently, the serum calcium-to-phosphorous (Ca/P) ratio revealed a disrupted regulation in the PD-Spl group. Moreover, since the hepatic function was normal, a further parameter for bone health is ALP blood concentrations. Many of the PD groups, including casein, showed higher levels than the normal range, but, again, the rats that were fed spirulina showed the highest value (Table 3). Altogether, blood analyses suggested a normal metabolic profile with a disrupted bone-related profile for the PD-Soy, PD-Spl, and PD-CP/I.

### 3.4. The Effect of the Alternative Protein Sources on Bone Quality

Next, we examined the properties of the “whole bone”. Some of the critical properties encompassed by the term bone quality are trabecular architecture, cortical geometry, and mechanical integrity [40]. Micro-CT and three-point bending techniques were used to estimate these parameters.

Analysis of the femora trabecular bone revealed that PD-CP/F had 1.16 folds larger BV/TV compared with that of the PD-Ctrl group. Opposingly, PD-Soy presented statistically lower BV/TV (Table 4). Due to the different growth patterns of the spine compared to the limbs, and different mechanical loads applied to them in the body, the results of the trabecular analysis of the vertebra were not correlated to the trabecular compartment of the femur. Vertebra analysis revealed no differences in BV/TV values between all alternative protein groups and Ctrl groups (Table 4).

The cortical bone analysis showed that the alternative protein source diets resulted in not only short femur lengths but also thinner widths, as the values of the cortical thickness (Ct.Th) and the medullary area (Ma.Ar) and the total cross-sectional area (Tt.Ar) were found compatible with the groups’ growth patterns and femora lengths. Yet, PD-Spl was the only group that had a a low Ct.Ar/Tt.Ar ratio. Moreover, BMD, which reflects the mineralization level of the bones, was normal in all groups, despite lower levels in the rats consuming spirulina and fly larva diets (Table 4).

The skeleton plays a critical mechanical role in bearing functional loads, and failure to do so results in fractures; thus, it is valuable to evaluate the mechanical properties of the bones. We found that the reduction in dietary protein led to an unusual appearance of the force–displacement curve, which differs from the standard Ctrl’s curve (Figure 4). Both Ctrl groups revealed a similar slope (as well as similar Young’s modulus) that represents mineralized and stiff bone (Table 4), but PD-Ctrl’s femora had very little post-yield displacement portion, suggesting a more brittle bone (Figure 4).

The PD-Soy and PD-CP/I groups had the highest values of Young’s modulus, which represents the bone’s material stiffness. Among all PD groups, the PD-Spl group exhibited significantly lower strength parameters, which are defined as yield load and maximal load. Alternatively, its values of fracture load and energy-to-fracture, which represents whole-bone toughness, were similar to those of the remaining alternative protein groups, but lower than PD-Ctrl (Table 4).

### 3.5. The Amino Acid Analysis of the Tested Diets

One of the parameters for protein quality is the ratio between the “most limiting amino acid” and its requirement. This would indicate a first approximation of the efficiency of utilization in the body. Since the PD-Ctrl diet resulted in standard growth (Appendix A), its amino acid profile was used as a reference for the minimum essential amino acid requirements for the rats’ normal maturation. Amino acid analysis showed that the most significant amino acids deficiencies were found in the PD-Soy and PD-CP/I diets (methionine + cysteine), afterward PD-Spl (lysine), then PD-Fly (methionine + cysteine), and PD-CP/F (tryptophan), (Table 5).

### 3.6. Data Integration

In order to infer the quality of each tested protein source and the relevance of the research model for protein evaluation, we created a heatmap prediction tool that integrates all the data from our study as compared to the referenced PD-Ctrl diet. Among all of the results, growth parameters and eating patterns demonstrated a clear distinction of the groups into three major classes: casein groups, afterward PD-Fly and PD-CP/F groups, and, eventually, PD-Soy, PD-Spl, and PD-CP/I (Figure 5A). This separation was also shown for bone parameters, even though the variation was less uniform (Figure 5B). Conversely, results of serum biochemistry analysis did not, show these trends (Figure 5B), while amino acid profiles confirmed heavier deficiencies in PD-Soy, PD-Spl and PD-CP/I diets than those of PD-Fly and PD-CP/F (Figure 5C).

## 4. Discussion

By reason of socio-demographic changes, accompanied by environmental damage and ethical problems, a transition to consumption of different and alternative protein sources in the human diet must occur [7]. In this research, we established a model that examines the health effects of these proteins that are soon going to be deep-rooted in people’s nutrition all over the world. We shed a light on the importance of a high-quality protein source in the daily diet; and revealed that although limited consumption of optimum protein, such as casein, has some negative effects on the bones’ mechanical strength—the body’s growth and development show no damage. This research uncovered that in a limited diet during the post-natal growth period, not one of the alternative protein sources has been found to be alimentary and efficient as casein. Yet, among them, fly larvae followed by chickpea flour are preferred. To elucidate the foundation for the different effects of the protein on the rats’ growth and skeletal quality, each protein has been investigated separately.

First, we examined what seemed to be the most surprising outcome of our research—the poor effect of soy protein consumption on growth and bone development. The experiment showed that a soy protein diet during a rats’ growth period led to reduced weight gain, retardation in the longitudinal growth, and modified bone strength parameters. Despite its definition as a complete protein, the low weight gain has also been demonstrated in previous works. Madani et.al found that after 28 experimental days, rats that were fed 10%, 20%, or 30% dietary casein gained more weight than rats that were fed the same amounts of soy protein (88 g, 184 g, 197 g compared with 29 g, 114 g, 144 g, respectively) [41]. The investigators suggested that this resulted from impaired amino-acid contents (especially lysine and methionine) in soybean protein. In this case, the protein underwent a breakdown and its amino acids were used for energy production. Likewise, this explanation goes well with the high serum urea level of our PD-Soy group.

Even though many studies have presented the potential effect of soy’s isoflavones in preventing menopause-induced osteoporotic bone loss [42], and despite the fact that it is used in baby formula, our study indicates that consumption of soy as the main protein source during a rats’ growth period leads to a severe disturbance in body and femur elongation, as well as the femur’s low trabecular volume (BV/TV). The beneficial effect of soy’s isoflavones on bone health was not observed.

However, Ahn and Park found that a daily isoflavone supplementation of up to 10.75 mg for growing rats stimulated longitudinal bone growth, and improved BMD and structural parameters [43]. Yet, it is important to bear in mind that our soy group consumed a limited amount of soy protein in the diet, and so a limited amount of isoflavones (an average of 0.55–0.93 mg of dietary isoflavones/day [44]). Moreover, it must be considered that soy products could contain anti-nutritional factors with toxic and/or adverse effects, such as protease inhibitors, lectins, saponins, tannins, phytate, etc. [45,46]. Due to that, it was suggested that high intake or prolonged consumption of soy protein could be injurious to health [45]. Taken together, our results have led us to the conclusion that daily and high consumption of soy protein during childhood and puberty could not exclusively support growth potential unless its natural composition has been supplemented and/or unless it underwent industrial nutritional changes. 

Spirulina is another alternative protein source we chose to examine in our study. Currently, it can be found in health food stores and is sold mainly in the form of health drinks, powders, or tablets [23] with a recommended daily dose of around 3–9 g [47]. Although its verified health effects as a food supplement, our results show that a spirulina diet caused impaired growth patterns, a shorter femur and lumbar spine, reduced mechanical strength, relatively low femoral BMD, and disruption in blood calcium-phosphorous homeostasis. These effects of spirulina have not been described before, and controvert the competence of spirulina to constitute a protein source that meets the body’s requirements. In their research, Fournier et al. showed that spirulina can replace half of the amount of casein in young rats fed a low-protein diet (10% protein), so it prevented a low-protein diet-induced bone and exhibited higher bone biomechanical properties [48]. In another study, growing rats that were fed different ratios of casein and spirulina showed that spirulina increased femur and lumbar spine length, improved bone strength and bone mineral content, caused a higher concentration of free calcium and phosphate in the blood, and stimulated growth-regulating hormone secretion [49]. 

One possible explanation for these contradictions might be the varied essential amino acid index (EAAI) of spirulina when different pH and temperature conditions are applied. For example, at 25 °C, the EAAI was only 0.4, whereas, at 30 °C, it rose up to 1.0 [50]. Variations in the nutrient profile might also occur under cultivation conditions [51] so different spirulina cultivations can have different nutritional values and therefore varied effects on the body. Second, it should be noted that our research is the only, and the first, that investigated spirulina as a sole dietary protein source, while the relative amount of protein in the diet did not exceed the minimum requirement. Since 10% protein in the diet should be sufficient when a high-quality protein source is supplied, our conclusion is that spirulina by itself is not an adequate protein source. Still, when it comes alongside with other proteins, spirulina’s good qualities are showing up [27].

Two chickpea forms have been examined in our work: chickpea flour (Kabuli type, ground without heat treatment), and chickpea isolate (a non-heated protein powder from isolated Kabuli chickpeas, which consists of a specific formulation of several protein fractions). Results show that although both sources lead to decreased growth performances and low bone mechanical strength, chickpea flour supported growth better than chickpea isolate. These unanticipated findings might suggest a loss of nutritive substances in the isolation process. [52]

In their research, Tavano et al. found that PER values of isolated unheated chickpea protein fractions (albumin, globulin, and glutelin) were significantly higher than that of unheated chickpea flour [53]. In contrast, our research’s PER calculations showed a higher value for chickpea flour than its isolated form, although we used unheated chickpea flour. This contradiction might be solved by the increased refusal of Tavano et al.’s rats to eat their unheated flour diet, for unknown reasons. However, their heated flour diet group presented a higher PER value than the unheated protein fraction. Furthermore, both Tavano et.al’s heated flour and our unheated flour had similar values of corrected PER (1.69 and 1.61, respectively). Thus, alongside previous works, we concluded that a diet based on natural raw materials is superior to a diet that incorporates individual nutrients that had undergone processing, isolation, and nutritional manipulations. 

Last but not least, we examine insects as a potential alternative protein source. Even though about a quarter of the human population consumes insects as common practice within their cultures, very limited research has been conducted to test their health effects. In our study, we investigated the efficiency of protein powder that was made from Mediterranean fruit fly (*Ceratitis capitata*) larvae, one of the potential insect species in the alternative proteins market. Our results showed that although a diet based on fly larvae protein did not bring into maximal growth patterns and standard bone mechanical properties; this protein is superior to the rest of the tested alternative sources. 

The data regarding different larvae species (*Tenebrio molitor* mealworm), show mixed effects. Gessner et al. showed that the replacement of casein with mealworm larvae exerts a pronounced reduction of liver and plasma lipid concentrations in a rat model for metabolic syndrome [54]. While Poelaert et al. demonstrated that the growth performance of rats that were fed mealworm larvae was negatively affected [55]. The investigators speculate that this reduced growth is a result of sulfur-containing amino acids deficiency compared with the rats’ requirement. In addition, the presence of chitin in this protein source may reduce amino acid digestibility and therefore decrease anabolic growth [56]. Our findings, together with previous findings, enhance the understanding regarding insects’ larvae as a potential appropriate protein source after the removal of chitin. Nevertheless, further comprehensive investigation is needed.

In this study, we created a protein quality evaluation model that sees protein sources not only as an amino acid reservoir to meet the body’s anabolic needs; but also, as a whole food source that contains other components and nutritional factors that could affect human growth and health. These different food contents also explain the fact that some of our findings were quite unexpected according to classic protein quality literature. One of the biggest challenges of nutritional research is to evaluate the value of a whole food source, although it may have been grown and processed differently around the world. Each of the aspects: crops’ management, preparation method, and chemical manipulation, determine the quality of the final protein produced prior to consumption [57]. Additionally, several other aspects of protein processing and nutrition, such as digestibility, bioaccessibility, conversion to muscle, and palatability, should be considered. This situation forces the scientific world to work harder and faster in assessing the quality and the health consequences of novel food sources, as well as the mixture of different sources of protein. This is especially important when it comes to alternative protein sources that humans will rely on as the main protein sources for future generations.

After all, the success of dealing with global warming and the increasing demand for protein will depend on “proving food safety, production costs, nutritional qualities, scalability and consumer acceptance” [58]. Protein safety and sustainability require evaluation and assessment by various methods [7]. Hence, our present research has important implications in the race for sustainability, as well as maintaining the health of the future population. 

## Figures and Tables

**Figure 1 nutrients-14-03769-f001:**
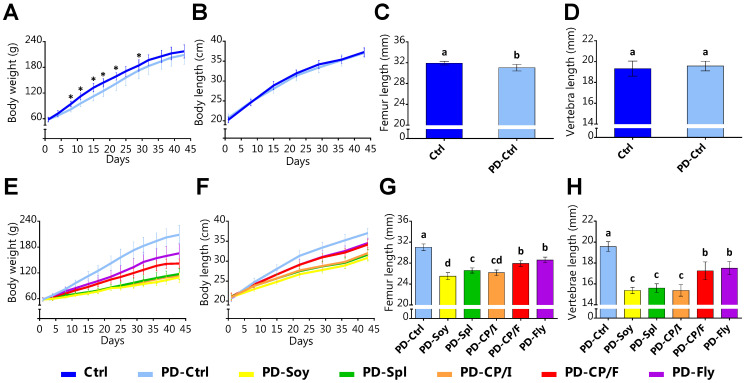
Growth patterns. (**A**) Ctrl groups’ body weight (g) throughout the experiment. (**B**) Ctrl groups’ body length (cm) throughout the experiment. (**C**) Ctrl groups’ right femur length measured by micro-CT (mm). (**D**) Ctrl groups’ 3rd–5th lumbar vertebra length measured by micro-CT (mm). (**E**) PD groups’ body weight (g) throughout the experiment. (**F**) PD groups’ body length (cm) throughout the experiment. (**G**) PD groups’ right femur length measured by micro-CT (mm). (**H**) PD groups’ 3rd–5th lumbar vertebra length measured by micro-CT (mm). Values are expressed as mean ± SD of *n* = 8 rats/group, different superscript letters are significantly different (*p* < 0.05) by one-way ANOVA followed by Tukey’s test.

**Figure 2 nutrients-14-03769-f002:**
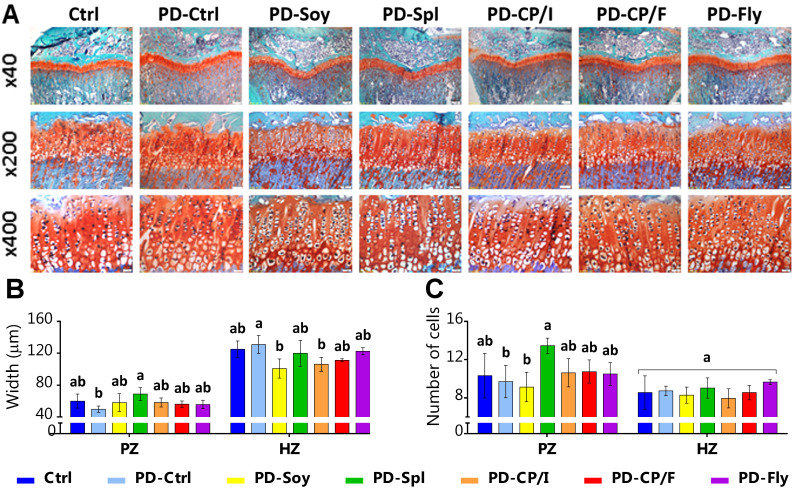
Histological evaluation of the tibial growth plates of 9-week-old rats. Transverse tissue sections of 5 μm were prepared by microtome. GP width and number of cells have been measured in cell A software at 10 selected locations. (**A**) Safranin-O staining of the tibial GP. (**B**) GP zones’ width (μm). (**C**) GP zones’ number of cells. GP, growth plate; PZ, proliferative zone; HZ, hypertrophic zone. Values are expressed as mean ± SD of *n* = 4 samples/group, different superscript letters are significantly different (*p* < 0.05) by one-way ANOVA followed by Tukey’s test.

**Figure 3 nutrients-14-03769-f003:**
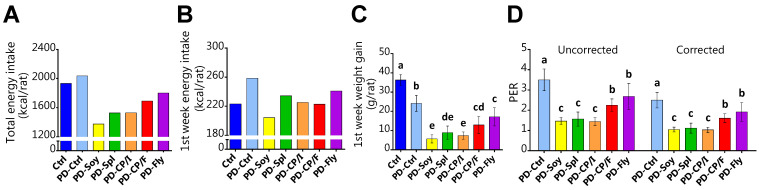
Energy consumption and utilization. (**A**) Total energy intake (kcal\rat). (**B**) 1st week total energy intake (kcal/rat). (**C**) 1st week total weight gain (g\rat). (**D**) Protein Efficiency Ratio (PER). Relative weight gain (g) per protein intake (g). PER examined under standardized conditions of weaning rats that were fed 10% protein in the diet for a test period of 4 weeks. Corrected PER values are standardized for PD-Cas = 2.5. Values are expressed as mean ± SD of *n* = 8 rats/group, different superscript letters are significantly different (*p* < 0.05) by one-way ANOVA followed by Tukey’s test.

**Figure 4 nutrients-14-03769-f004:**
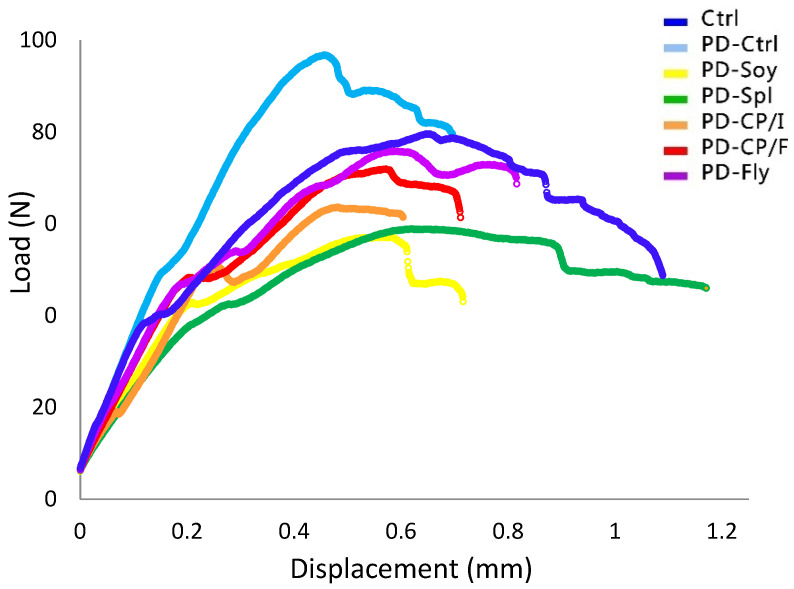
Force–Displacement curves of experimental groups. Mechanical properties were evaluated using a three-point bending experiment performed on the bones of all rats from the seven groups.

**Figure 5 nutrients-14-03769-f005:**
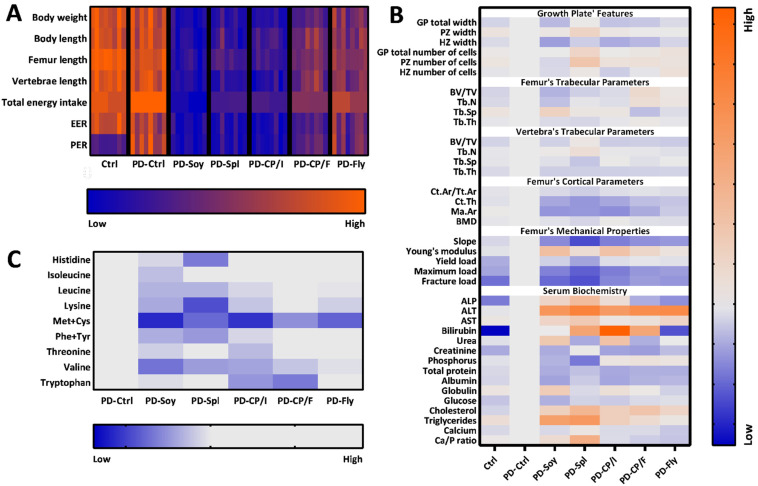
Evaluating protein quality through growth and eating patterns, bone features, serum parameters, and amino acid composition. (**A**) Heatmap representing the rats’ growth and eating performances. (**B**) Heatmap representing groups’ bone analyses’ results and serum biochemistry relative to PD-Ctrl’ values. (**C**) Heatmap representing protein sources’ amino acid content relative to PD-Ctrl’ AA profile.

**Table 1 nutrients-14-03769-t001:** Ingredient Composition and Nutritional Content of Diets. Values of diet composition are presented as grams per dry weight.

	Ctrl	PD-Ctrl	PD-Soy	PD-Spl	PD-CP/I	PD-CP/F	PD-Fly
Diet Composition	g/kg	g/950 g	g/956 g	g/923 g	g/949 g	g/978 g	g/962 g
Cornstarch	397	406	406	375	401	235	397
Dextrinized cornstarch	132	135	135	125	133	78	132
Sucrose	100	102	102	94	101	59	100
Soybean oil	70	110	111	107	111	84	97
Cellulose fibers	50	50	50	19	50	0	50
Casein (≥85% protein)	200	98					
L-Cystine	3	1					
Soy protein isolate			103				
Spirulina powder				156			
Chickpea protein isolate					104		
Chickpea flour						475	
Fly larvae protein powder							139
Mineral mix (AIN-93G-MX)	35	35	35	35	35	35	35
Vitamin mix (AIN-93G-MX)	10	10	10	10	10	10	10
Choline bitartrate	2.5	2.5	2.5	2.5	2.5	2.5	2.5
Tert-Butylhydroquinone	0.014	0.014	0.014	0.014	0.014	0.014	0.014
Nutritional level							
Energy, kcal/g	3.10	3.48	3.42	3.62	3.60	3.46	3.40
Protein, %	21	10	10	10	10	10	10
Carbohydrate, %	61	62	62	62	62	62	62
Fat, %	18	28	28	28	28	28	28
Fibers, g/kg	50	52.6	52.4	54.1	52.7	55.2	51.9

**Table 2 nutrients-14-03769-t002:** Protein Sources Composition. Values are presented as kcal per dry weight and as grams per dry weight. Caseinate (MEGGELE), Soy protein isolate (SINOPRO), Spirulina powder (Abundance), Chickpea protein isolate (ChickP), Chickpea flour (Refresh), Fly larvae protein powder (Flying Spark).

100 gr Protein Source	Caseinate	Soy Isolate	Spirulina Powder	Chickpea Isolate	Chickpea Flour	Fly Larvae Powder
Energy (kcal)	375	365	380	364	356	319
Protein (g)	91	90	60	90	20	67
Carbohydrate (g)	0.1	0	28	7	50	10
Fat (g)	0.8	0.5	3	0	6	10
Fibers (g)	0	0	20	0	11.5	0

**Table 3 nutrients-14-03769-t003:** Serum biochemistry analysis. ALP, Alkaline Phosphatase; ALT, Alanine Transaminase; AST, Aspartate Aminotransferase. Values are expressed as mean ± SD of *n* = 8 rat/group, different superscript letters are significantly different (*p* < 0.05) by one-way ANOVA followed by Tukey’s test. Bold numbers significantly differ between diets and from the normal range.

	Blood Parameter	Normal Range	Ctrl	PD-Ctrl	PD-Soy	PD-Spl	PD-CP/I	PD-CP/F	PD-Fly
Liver function	ALT (U/L)	30–82	23 ± 5 ^b^	23 ± 4 ^b^	44 ± 11 ^a^	49 ± 8 ^a^	43 ± 6 ^a^	46 ± 9 ^a^	47 ± 12 ^a^
	AST (U/L)	70–178	101 ± 32 ^ab^	96 ± 14 ^b^	117 ± 24 ^ab^	129 ± 14 ^a^	102 ± 13 ^ab^	106 ± 10 ^ab^	117 ± 25 ^ab^
	Bilirubin (mg/dL)	0.04–0.21	0.01 ± 0.01 ^c^	0.03 ± 0.00 ^bc^	0.03 ± 0.02 ^bc^	0.05 ± 0.03 ^ab^	0.06 ± 0.02 ^a^	0.04 ± 0.03 ^ab^	0.01 ± 0.02 ^bc^
Renal function	Urea (mg/dL)	28.8–61.3	36.3 ± 3.0 ^b^	37.5 ± 8.0 ^b^	50.8 ± 6.2 ^a^	30.3 ± 6.8 ^b^	53.0 ± 10.8 ^a^	30.8 ± 1.7 ^b^	37.4 ± 6.7 ^b^
	Creatinine (mg/dL)	0.26–0.65	0.37 ± 0.07 ^ab^	0.46 ± 0.05 ^a^	0.37 ± 0.05 ^ab^	0.46 ± 0.09 ^a^	0.36 ± 0.06 ^ab^	0.35 ± 0.05 ^b^	0.39 ± 0.06 ^ab^
Nutrtional status	Albumin (g/dL)	3.5–5.1	5.1 ± 0.4 ^ab^	5.5 ± 0.5 ^a^	4.3 ± 0.6 ^c^	4.9 ± 0.5 ^abc^	4.3 ± 0.3 ^c^	4.6 ± 0.6 ^bc^	4.6 ± 0.4 ^bc^
	Total protein (g/dL)	**6–7.3**	**6.1 ± 0.8 ^ab^**	**6.5 ± 0.8 ^a^**	**5.2 ± 0.7 ^b^**	**5.6 ± 0.7 ^ab^**	**5.2 ± 0.5 ^b^**	**5.3 ± 0.6 ^b^**	**5.3 ± 0.6 ^b^**
MetabolicParameters	Glucose (mg/dL)	113–185	193 ± 32 ^a^	217 ± 30 ^a^	180 ± 27 ^a^	202 ± 28 ^a^	196 ± 29 ^a^	206 ± 19 ^a^	209 ± 34 ^a^
	Cholesterol (mg/dL)	71–148	77 ± 12 ^c^	83 ± 11 ^bc^	105 ± 19 ^abc^	127 ± 22 ^a^	106 ± 15 ^ab^	115 ± 23 ^a^	101 ± 17 ^abc^
	Triglycerides (mg/dL)	16–77	46 ± 16 ^bc^	41 ± 14 ^c^	74 ± 21 ^ab^	77 ± 25 ^a^	53 ± 20 ^abc^	48 ± 7 ^bc^	43 ± 11 ^c^
Bone parameters	Calcium (mg/dL)	10.1–12	10.9 ± 1.4 ^a^	11.6 ± 1.4 ^a^	11.0 ± 1.2 ^a^	12.3 ± 1.2 ^a^	11.0 ± 1.1 ^a^	11.0 ± 1.5 ^a^	10.4 ± 0.9 ^a^
	Phosphorus (mg/dL)	7.1–11.6	7.4 ± 0.9 ^ab^	7.6 ± 0.5 ^ab^	6.3 ± 1.2 ^bc^	5.0 ± 0.7 ^c^	8.0 ± 1.1 ^ab^	8.3 ± 1.4 ^a^	8.0 ± 0.8 ^ab^
	Ca/P ratio	-	**1.5 ± 0.2 ^bc^**	**1.4 ± 0.1 ^bc^**	**1.7 ± 0.1 ^b^**	**2.4 ± 0.3 ^a^**	**1.4 ± 0.2 ^bc^**	**1.3 ± 0.1 ^c^**	**1.3 ± 0.1 ^c^**
	ALP (U/L)	**161.2–258**	**201.6 ± 34.6 ^d^**	**302.7 ± 56.7 ^bc^ **	**371.8 ± 52.3 ^ab^ **	**448.2 ± 97.5 ^a^ **	**339.5 ± 33.3 ^b^ **	**247.2 ± 41.0 ^cd^ **	**220.8 ± 57.5 ^cd^ **

**Table 4 nutrients-14-03769-t004:** Morphometric and mechanic characteristics. Bones were scanned by Micro-CT to determine geometric parameters. After reconstructing 2D and 3D analyses were performed. Trabecular bone parameters: Bone volume over total volume, BV/TV (%); Trabecular number, Tb.N (1/mm); Trabecular separation, Tb.Sp (mm); Trabecular thickness, Tb.Th (mm). Cortical bone parameters: Total Cross-Sectional Area, Tt.Ar (mm^2^); Cortical Bone Area, Ct.Ar (mm^2^); Cortical area fraction, Ct.Ar/Tt.Ar (%); Cortical thickness, Ct.Th (mm); Medullary area, Ma.Ar (mm^2^); Bone mineral density, BMD (g/cm^3^). Mechanical properties were evaluated using a three-point bending experiment. Biomechanical parameters obtained from load–displacement curve: Slope (N/mm); Young’s modulus (N/mm^2^); Yield load (N); Max load (N); Fracture load (N); Energy to fracture, E to F (N × mm). Values are expressed as mean ± SD of *n* = 8 rats/group, different superscript letters are significantly different (*p* < 0.05) by one-way ANOVA followed by Tukey’s test.

	Ctrl	PD-Ctrl	PD-Soy	PD-Spl	PD-CP/I	PD-CP/F	PD-Fly
Trabecular analysis of the femur							
BV/TV (%)	33.98 ± 2.21 ^bc^	36.52 ± 2.24 ^b^	30.48 ± 3.08 ^c^	33.50 ± 3.95 ^bc^	34.78 ± 4.93 ^bc^	42.58 ± 3.24 ^a^	38.11 ± 4.27 ^ab^
Tb.N (1/mm)	3.10 ± 0.15 ^bc^	3.28 ± 0.17 ^abc^	2.87 ± 0.25 ^c^	3.21 ± 0.37 ^bc^	3.16 ± 0.41 ^bc^	3.72 ± 0.30 ^a^	3.48 ± 0.29 ^ab^
Tb.Sp (mm)	0.24 ± 0.01 ^ab^	0.22 ± 0.01 ^bc^	0.28 ± 0.03 ^a^	0.23 ± 0.03 ^bc^	0.23 ± 0.03 ^bc^	0.19 ± 0.03 ^c^	0.21 ± 0.02 ^bc^
Tb.Th (mm)	0.109 ± 0.004 ^abc^	0.111 ± 0.002 ^ab^	0.105 ± 0.003 ^bc^	0.104 ± 0.003 ^c^	0.109 ± 0.003 ^abc^	0.114 ± 0.005 ^a^	0.109 ± 0.005 ^abc^
Trabecular analysis of the vertebra							
BV/TV (%)	29.05 ± 1.86 ^ab^	31.17 ± 1.44 ^ab^	28.85 ± 1.63 ^b^	32.19 ± 2.18 ^a^	28.28 ± 2.30 ^b^	28.44 ± 1.99 ^b^	28.16 ± 2.95 ^b^
Tb.N (1/mm)	2.44 ± 0.12 ^b^	2.48 ± 0.08 ^b^	2.51 ± 0.12 ^b^	2.79 ± 0.15 ^a^	2.45 ± 0.18 ^b^	2.44 ± 0.16 ^b^	2.39 ± 0.16 ^b^
Tb.Sp (mm)	0.306 ± 0.015 ^a^	0.311 ± 0.017 ^a^	0.301 ± 0.015 ^ab^	0.276 ± 0.017 ^b^	0.314 ± 0.024 ^a^	0.302 ± 0.019 ^ab^	0.310 ± 0.022 ^a^
Tb.Th (mm)	0.118 ± 0.003 ^b^	0.125 ± 0.003 ^a^	0.114 ± 0.003 ^b^	0.115 ± 0.002 ^b^	0.115 ± 0.002 ^b^	0.116 ± 0.001 ^b^	0.117 ± 0.005 ^b^
Cortical analysis of the femur							
Tt.Ar (mm^2^)	9.78 ± 0.68 ^a^	9.91 ± 0.91 ^a^	7.01 ± 0.41 ^cd^	6.82 ± 0.41 ^d^	6.86 ± 0.40 ^d^	7.83 ± 0.32 ^bc^	8.61 ± 0.49 ^b^
Ct.Ar (mm^2^)	4.93 ± 0.25 ^a^	5.11 ± 0.30 ^a^	3.44 ± 0.25 ^c^	3.21 ± 0.15 ^c^	3.41 ± 0.16 ^c^	3.94 ± 0.14 ^b^	4.26 ± 0.20 ^b^
Ct.Ar/Tt.Ar(%)	50.60 ± 2.52 ^a^	51.71 ± 2.90 ^a^	49.01 ± 1.12 ^ab^	47.16 ± 1.70 ^b^	49.74 ± 1.60 ^ab^	50.31 ± 1.70 ^ab^	49.52 ± 2.26 ^ab^
Ct.Th (mm)	0.47 ± 0.02 ^a^	0.49 ± 0.02 ^a^	0.39 ± 0.01 ^c^	0.36 ± 0.01 ^c^	0.39 ± 0.01 ^c^	0.42 ± 0.01 ^b^	0.43 ± 0.01 ^b^
Ma.Ar (mm^2^)	4.84 ± 0.53 ^a^	4.80 ± 0.68 ^a^	3.57 ± 0.17 ^c^	3.60 ± 0.30 ^c^	3.45 ± 0.28 ^c^	3.89 ± 0.25 ^bc^	4.35 ± 0.40 ^ab^
BMD (g/cm^3^)	1.34 ± 0.03 ^abc^	1.36 ± 0.02 ^ab^	1.32 ± 0.02 ^bcd^	1.27 ± 0.04 ^d^	1.37 ± 0.03 ^a^	1.31 ± 0.029 ^bcd^	1.308 ± 0.018 ^cd^
Mechanical properties of the left femora							
Slope (N/mm)	278.92 ± 44.5 ^a^	296.62 ± 40.4 ^a^	211.07 ± 71.4 ^b^	153.88 ± 17.4 ^c^	199.76 ± 25.8 ^bc^	215.19 ± 35.4 ^b^	222.18 ± 26.2 ^b^
Young’s modulus (N/mm^2^)	1161.79 ± 203.4 ^c^	1194.03 ± 129.6 ^c^	1727.27 ± 113.4 ^a^	1368.23 ± 225.2 ^c^	1693.95 ± 232.8 ^ab^	1430.46 ± 141.4 ^bc^	1283.57 ± 224.9 ^c^
Yield load (N)	37.19 ± 4.7 ^bc^	45.70 ± 3.0 ^a^	42.03 ± 3.6 ^ab^	36.04 ± 2.0 ^c^	43.92 ± 4.3 ^a^	45.01 ± 3.8 ^a^	44.41 ± 4.3 ^a^
Max load (N)	73.69 ± 6.4 ^b^	92.86 ± 8.7 ^a^	63.43 ± 4.8 ^c^	53.86 ± 3.0 ^d^	62.17 ± 3.4 ^c^	69.25 ± 3.8 ^bc^	71.41 ± 3.5 ^b^
Fracture load (N)	49.33 ± 14.6 ^bc^	77.95 ± 9.0 ^a^	47.57 ± 15.6 ^bc^	41.91 ± 10.6 ^c^	54.47 ± 10.5 ^bc^	59.63 ± 5.0 ^b^	57.98 ± 6.0 ^bc^
E to F (N×mm)	72.87 ± 12.6 ^a^	57.22 ± 12.5 ^ab^	35.13 ± 10.2 ^c^	41.23 ± 20.5 ^bc^	28.53 ± 7.1 ^c^	37.69 ± 9.7 ^c^	47.57 ± 11.6 ^bc^

**Table 5 nutrients-14-03769-t005:** Diets’ essential amino acids content as a percentage of PD-Ctrl diet’s amino acids profile. Part of the sulfuric amino acid methionine can be replaced by cysteine, and part of the aromatic amino acid phenylalanine can be replaced by tyrosine. The most limiting amino acid in every diet is underlined.

	PD-Ctrl Diet	PD-Soy Diet	PD-Spl Diet	PD-CP/I Diet	PD-CP/F Diet	PD-Fly Diet
Histidine	100%	94%	67%	103%	190%	109%
Isoleucine	100%	87%	109%	104%	113%	120%
Leucine	100%	84%	84%	94%	108%	98%
Lysine	100%	81%	54%	89%	123%	92%
Methionine + Cysteine	100%	43%	62%	46%	73%	60%
Phenylalanine + Tyrosine	100%	82%	76%	94%	181%	122%
Threonine	100%	92%	111%	86%	119%	124%
Valine	100%	65%	77%	79%	89%	97%
Tryptophan	100%	96%	128%	75%	67%	148%

## Data Availability

Not applicable.

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
