# Peer review of "The Use of Post-Natal Skeleton Development as Sensitive Preclinical Model to Test the Quality of Alternative Protein Sources in the Diet"

_nutrients, 2022, doi:10.3390/nu14183769_

Round 1
Reviewer 1 Report
Nutrients 1876751. The Use of Post-Natal Skeletn Development as Sensitive Pre-clinical Model to Test the Quality of Alternative Protein Sources in the Diet
The authors of this proposal have evaluated different sources of protein and their impact on the development and health of the bone in an animal model.
Appreciable authors,
Through the review of the manuscript, I can appreciate the effort in the quality of the research. However, it is essential to make some pressure and improvement within your manuscript; in this sense, allow me to request some changes. Authors must review in detail the instructions to the authors. In the text, reference numbers should be placed in square brackets [ ] and placed before the punctuation; for example, [1], [1–3], or [1,3]. https://www.mdpi.com/journal/nutrients/instructions.
References should be described as described, the author's instructions depending on the type of work, which you can see in the following link https://www.mdpi.com/journal/nutrients/instructions.
Additionally, some changes are necessary at the version of the manuscript, please find you highlighted text in the PDF file attached.
Other minor requirements are of importance to support the manuscript:
Line 2
It says: Skeletn
It should say: Skeleton
Line: 15
It says: necessary
Maybe it should say: essential
Line 35
It says: synthesis
Maybe it should say: the synthesis
Line 40
It says: One of the many organs that require a sufficient dietary protein intake is the musculoskeletal system
Maybe it should say: The musculoskeletal system is one of the many organs that require a sufficient dietary protein intake
Line 50
It says: to conversion
It should say: to the conversion
Line 53
It says: the conventional
It should say: conventional
Line 61
It says: [11], [12].
It should say: [11,12].
Line 62
It says: that soy contains
It should say: that soy contains,
Line 64
It says: [13], [14].
It should say: [13, 14].
Line 71
It says: [18], [20].
It should say: [18, 20].
Line 73
It says: a blue-green cyanobacteria
It should say: blue-green cyanobacteria
Line 73
It says: spirulina listed
It should say: spirulina is listed
Line 75
It says: been also
It should say: also been
Line 80-82
It says: In regards to the skeletal system, a previous study revealed that an addition of 5 g/kg spirulina to a low-protein-low-micronutrients-diet for 81 growing rats improved their growth performances and bone quality
It should say: Regarding the skeletal system, a previous study revealed that adding 5 g/kg spirulina to a low-protein-low-micronutrients-diet for 81 growing rats improved their growth performances and bone quality [27].
Line 83, 84
It says: Insects are part of the traditional diets of at least 2 billion people all over the world, and particularly in parts of Asia, Africa and South America [7].
It should say: Insects are part of the traditional diets of at least 2 billion people worldwide, particularly in parts of Asia, Africa, and South America [7].
Line 89
It says: an adequate
It should say: a good
Line 89
It says: mono unsaturated
It should say: monounsaturated
Line 90
It says: such iron
It should say: such as iron
Line 91
It says: and health
It should say: and the health
Line 94
It says: In order to
It should say: To do so
Line 95
It says: young rats at their post
It should say: young rats during their post
Line 98
It says: sources as
It should say: sources such as
Line 117
It says: Tibias were fixed for histological analysis.
It should say: Finally, tibias were fixed for histological analysis.
Line 142
It says: [30], [31].
It should say: [30, 31].
Line 144
It says: xylen
It should say: xylene
Line 156
It says: [32], [37].
It should say: [32, 37].
Line 173
It says: [34], [35].
It should say: [34, 35].
Author Response
We would like to thank you and the reviewers for the candid review of our manuscript “The Use of Post-Natal Skeleton Development as Sensitive Pre-clinical Model to Test the Quality of Alternative Protein Sources in the Diet” (Nutrients 1876751). We read carefully all the reviewers’ comments and addressed all of them as listed below. We feel that these modifications improved the quality of the manuscript, and hope you will find it suitable for publication.
Reviewer 1:
The authors of this proposal have evaluated different sources of protein and their impact on the development and health of the bone in an animal model.
Through the review of the manuscript, I can appreciate the effort in the quality of the research. However, it is essential to make some pressure and improvement within your manuscript; in this sense, allow me to request some changes. Authors must review in detail the instructions to the authors.
Thanks a lot for the suggestions.
In the text, reference numbers should be placed in square brackets [ ] and placed before the punctuation; for example, [1], [1–3], or [1,3].
https://www.mdpi.com/journal/nutrients/instructions. References should be described as described, the author's instructions depending on the type of work, which you can see in the following link https://www.mdpi.com/journal/nutrients/instructions.
All the reference were modified according to the instructions, both in the text and in references list.
Additionally, some changes are necessary at the version of the manuscript, please find you highlighted text in the PDF file attached.
Modified
Other minor requirements are of importance to support the manuscript:
Line 2 It says: Skeletn It should say: Skeleton
Corrected
Line: 15 It says: necessary Maybe it should say: essential
Corrected
Line 35 It says: synthesis Maybe it should say: the synthesis
Corrected
Line 40
It says: One of the many organs that require a sufficient dietary protein intake is the musculoskeletal system
Maybe it should say: The musculoskeletal system is one of the many organs that require a sufficient dietary protein intake
Corrected
Line 50 It says: to conversion It should say: to the conversion
Corrected
Line 53 It says: the conventional It should say: conventional
Corrected
Line 61 It says: [11], [12]. It should say: [11,12].
Corrected
Line 62 It says: that soy contains It should say: that soy contains,
Corrected
Line 64 It says: [13], [14]. It should say: [13, 14].
Corrected
Line 71 It says: [18], [20]. It should say: [18, 20].
Corrected
Line 73 It says: a blue-green cyanobacteria It should say: blue-green cyanobacteria
Corrected
Line 73 It says: spirulina listed It should say: spirulina is listed
Corrected
Line 75 It says: been also It should say: also been
Corrected
Line 80-82
It says: In regards to the skeletal system, a previous study revealed that an addition of 5 g/kg spirulina to a low-protein-low-micronutrients-diet for 81 growing rats improved their growth performances and bone quality
It should say: Regarding the skeletal system, a previous study revealed that adding 5 g/kg spirulina to a low-protein-low-micronutrients-diet for 81 growing rats improved their growth performances and bone quality [27].
Corrected
Line 83, 84
It says: Insects are part of the traditional diets of at least 2 billion people all over the world, and particularly in parts of Asia, Africa and South America [7].
It should say: Insects are part of the traditional diets of at least 2 billion people worldwide, particularly in parts of Asia, Africa, and South America [7].
Corrected
Line 89 It says: an adequate It should say: a good
Corrected
Line 89 It says: mono unsaturated It should say: monounsaturated
Corrected
Line 90 It says: such iron It should say: such as iron
Corrected
Line 91 It says: and health It should say: and the health
Corrected
Line 94 It says: In order to It should say: To do so
Corrected
Line 95 It says: young rats at their post It should say: young rats during their post
Corrected
Line 98 It says: sources as It should say: sources such as
Corrected
Line 117 It says: Tibias were fixed for histological analysis. It should say: Finally, tibias were fixed for histological analysis.
Corrected
Line 142 It says: [30], [31]. It should say: [30, 31].
Corrected
Line 144 It says: xylen It should say: xylene
Corrected
Line 156 It says: [32], [37]. It should say: [32, 37].
Corrected
Line 173 It says: [34], [35]. It should say: [34, 35]
Corrected
Reviewer 2 Report
Paper Nutrients-1876751
Title: The Use of Post-Natal Skeleton Development as Sensitive Preclinical Model to Test the Quality of Alternative Protein Sources in the Diet
The article proposed an interesting and complete evaluation of the quality of alternative protein sources in the diet of female rats. The study is objective, and the conception of the study was well done. I have some comments on points that need to be clarified and some suggestions that may improved the paper.
1. Title: There is a misspelling of the word “skeleton”. Please correct.
2. Line 16: Rewrite the expression “by reason of”.
3. Line 54: Yeast (like single cell protein) protein should also be in this list.
4. Line 55: Add if in wet or dry basis.
5. Line 77: Avoid using restrictive words such as “only”. There are other microalgae with this pigment. Please, rewrite.
6. Line 101: Why did the authors research solely on female rat subjects?
7. Line 201: What was included in the “complete biochemical analysis”?
8. Line 216: Were the normality and homoscedasticity of data checked, which are requirements to run an ANOVA and a Tukey-HSD tests? If so, how (by which tests)?
9. Table 3: There is a misspelling at the fifth line “Metabolic parameters”.
10. Table 3: Keep the same number of decimals for averages and standard deviations, all throughout the table, please. Don’t forget only to keep a pertinent number of decimals.
11. Line 371: The idea of the heatmap using all the data is interesting but no information is given in the Methods section regarding how the heatmaps were produced (methodology and software, etc.). Please clarify.
12. Line 406: There is a space between the numbers and the “g”.
13. Line 415: Instead of writing “the great effect”, be more specific.
14. Line 420: I noticed that the authors discuss a lot the effect of isoflavones. What was the exact amount of these compounds in the soy product given to the rats? Was it measured in this research?
15. Lines 421-423: Yes, this statement is true. That is why I do not understand how the statement given in lines 57-59 is also true. There are differences in the digestibility and thus the bioaccessibility of soy protein in comparison to animal sourced proteins. I think the “digestion” aspect of this discussion should be further explored.
16. Line 428: Not only nutritional changes, but processing to increase its digestibility and protein real absorption.
17. Line 435: Is “disrupted effects” the best description?
18. Lines 437-443: Indeed, the study would benefit from the study of mixed-origin proteins, with different proportions for proteins of each source. This possibility of research should be added in the perspectives of the work.
19. Lines 462-470: It is not clear why the authors discuss about the heated treatments for chickpea products if none of the products used in the work were heated.
20. Lines 478-481: This phrase should be written to avoid misconceptions. It is clear that the amino acid profile of this insect protein meets the composition of animal-based proteins. However, as this work elucidates, the amino acid profile is only one of the parameters that describe a protein that is considered “a complete protein source”. There are many other parameters that are considered. So I suggest the authors avoid writing it because it can be misinterpreted by readers that would think that this protein promotes the same muscular growth, etc. than caseins.
21. Line 495: … “that could affect…”. Please correct.
22. Line 496: “This… what?” Please avoid writing “this” without clearly specifying what. This result?
23. Lines 499-501: There are many other aspects of protein processing and nutrition that should be added here. For instance, protein digestibility, bioaccessibility and real absorption, conversion to muscle, palatability, acceptability, food safety, etc. This perspectives section should be improved cause new research projects could benefit from the authors’ insights and comments. What should be done in the following works? Which other parameters could be added to assess protein quality?
24. Lines 507-508: This phrase should be revised, English need to be improved.
Author Response
We would like to thank you and the reviewers for the candid review of our manuscript “The Use of Post-Natal Skeleton Development as Sensitive Pre-clinical Model to Test the Quality of Alternative Protein Sources in the Diet” (Nutrients 1876751). We read carefully all the reviewers’ comments and addressed all of them as listed below. We feel that these modifications improved the quality of the manuscript, and hope you will find it suitable for publication.
The article proposed an interesting and complete evaluation of the quality of alternative protein sources in the diet of female rats. The study is objective, and the conception of the study was well done. I have some comments on points that need to be clarified and some suggestions that may improve the paper.
Thanks for the important suggestions and discussion.
- Title: There is a misspelling of the word “skeleton”. Please correct. Corrected
- Line 16: Rewrite the expression “by reason of”. Corrected
- Line 54: Yeast (like single cell protein) protein should also be in this list. Added
- Line 55: Add if in wet or dry basis. Added
- Line 77: Avoid using restrictive words such as “only”. There are other microalgae with this pigment. Please, rewrite. Corrected
- Line 101: Why did the authors research solely on female rat subjects? We usually use female rats in our nutritional experiments, since bone metabolic disease such as osteoporosis is more abundant in females, making it even more crucial to optimize growth conditions to reach PBM in young age.
- Line 201: What was included in the “complete biochemical analysis”? All the parameters that were checked are detailed in the results, at table 3. Serum biochemistry analysis.
- Line 216: Were the of data checked, which are requiremetnts to run an ANOVA and a Tukey-HSD tests? If so, how (by which tests)? Normality and homoscedasticity test of the data before statistic was conducted by Bartlett’s test, this information was added to M&M.
- Table 3: There is a misspelling at the fifth line “Metabolic parameters”. Corrected
- Table 3: Keep the same number of decimals for averages and standard deviations, all throughout the table, please. Don’t forget only to keep a pertinent number of decimals. The table was corrected as suggested.
- Line 371: The idea of the heatmap using all the data is interesting but no information is given in the Methods section regarding how the heatmaps were produced (methodology and software, etc.). Please clarify. Description of the Heatmap preparation was added to material and method section.
- Line 406: There is a space between the numbers and the “g”. Corrected
- Line 415: Instead of writing “the great effect”, be more specific. Corrected
- Line 420: I noticed that the authors discuss a lot the effect of isoflavones. What was the exact amount of these compounds in the soy product given to the rats? Was it measured in this research? We did not check empirically the amount of isoflavones in our soy-diets. We discuss the issue based on the literature information regarding the amount of this ingredient in soy, and calculated the averaged amount to our diet. We further connect it to possible effect of the diet together with other known ingredients in soy. This subject is addressed in the discussion in lines 424-428.
- Lines 421-423: Yes, this statement is true. That is why I do not understand how the statement given in lines 57-59 is also true. There are differences in the digestibility and thus the bioaccessibility of soy protein in comparison to animal sourced proteins. I think the “digestion” aspect of this discussion should be further explored. The different between the two statements is based on different scores that are used to evaluate protein quality. The PDCASS score of soy products such as soy isolate or soy concentrate is the same as that of animal protein (according to the literature), and this is what stated in the introduction. In the discussion we refer to the surprising results we got, and suggest that the reason is that soy products that contain a lot of anti-nutritional factors that impair their nutritional quality. PDCAAS address protein quality based on the amino acids absorbed as compared to body's requirements for essential amino acids. PDCAAS does not refer to other components that may be in the product. The view is not on the whole food, but only on one component of it, which limit the ability to test the quality of the protein sources products. Additionally, soy isolates go through different processing, depending on their purpose. The degree of processing and cleanliness in the product also affects the content of anti-nutritional factors (soy isolate intended for baby formulas versus soy isolate intended for use by the food industry in order to create a certain texture or obtain certain percentages of protein in an ultra-processed product anyway).
- Line 428: Not only nutritional changes, but processing to increase its digestibility and protein real absorption. We agree and the phrase” “industrial nutritional changes” that we use here include of course processing to increase its digestibility and protein real absorption.
- Line 435: Is “disrupted effects” the best description? We erase disrupted.
- Lines 437-443: Indeed, the study would benefit from the study of mixed-origin proteins, with different proportions for proteins of each source. This possibility of research should be added in the perspectives of the work. This issue was added in line 512.
- Lines 462-470: It is not clear why the authors discuss about the heated treatments for chickpea products if none of the products used in the work were heated. This was further explained in the discussion, based on the literature and since processing of food (such as heating) has crucial effect on diet quality.
- Lines 478-481: This phrase should be written to avoid misconceptions. It is clear that the amino acid profile of this insect protein meets the composition of animal-based proteins. However, as this work elucidates, the amino acid profile is only one of the parameters that describe a protein that is considered “a complete protein source”. There are many other parameters that are considered. So I suggest the authors avoid writing it because it can be misinterpreted by readers that would think that this protein promotes the same muscular growth, etc. than caseins. We agree and remove the sentence to avoid misconception as suggested.
- Line 495: … “that could affect…”. Please correct. Corrected
- Line 496: “This… what?” Please avoid writing “this” without clearly specifying what. This result? Corrected to “These different food contents…”
- Lines 499-501: There are many other aspects of protein processing and nutrition that should be added here. For instance, protein digestibility, bioaccessibility and real absorption, conversion to muscle, palatability, acceptability, food safety, etc. This perspectives section should be improved cause new research projects could benefit from the authors’ insights and comments. What should be done in the following works? Which other parameters could be added to assess protein quality? Thanks, this aspect was also added in the paragraph lines 510-512.
- Lines 507-508: This phrase should be revised, English need to be improved. The sentence was revised to: “Proteins safety and sustainability require evaluation and assessment by various methods [7]”